Dispersal and metapopulation stability

Wang Shaopeng wangshop@gmail.com
Haegeman Bart
Loreau Michel
Centre for Biodiversity Theory and Modelling, Station d’Ecologie Expérimentale du CNRS , Moulis , France
Hastings Alan
Electronic publication date: 2015 Oct 1
Publication date: 2015
Volume: 3
Electronic Location ID: e1295
Received 2015 Jul 1; Accepted 2015 Sep 15
Copyright: © 2015 Wang et al.
Copyright year: 2015
Copyright holder: Wang et al.
License: This is an open access article distributed under the terms of the Creative Commons Attribution License, which permits unrestricted use, distribution, reproduction and adaptation in any medium and for any purpose provided that it is properly attributed. For attribution, the original author(s), title, publication source (PeerJ) and either DOI or URL of the article must be cited.
License URL: https://creativecommons.org/licenses/by/4.0/

Keywords: Asymmetry, Dispersal, Metapopulation, Variability, Synchrony, Stability, Corridor, Spatial heterogeneity

Funding: TULIP Laboratory of Excellence ANR-10-LABX-41 This work was supported by the TULIP Laboratory of Excellence (ANR-10-LABX-41). The funders had no role in study design, data collection and analysis, decision to publish, or preparation of the manuscript.

==============================
Metapopulation dynamics are jointly regulated by local and spatial factors. These factors may affect the dynamics of local populations and of the entire metapopulation differently. Previous studies have shown that dispersal can stabilize local populations; however, as dispersal also tends to increase spatial synchrony, its net effect on metapopulation stability has been controversial. Here we present a simple metapopulation model to study how dispersal, in interaction with other spatial and local processes, affects the temporal variability of metapopulations in a stochastic environment. Our results show that in homogeneous metapopulations, the local stabilizing and spatial synchronizing effects of dispersal cancel each other out, such that dispersal has no effect on metapopulation variability. This result is robust to moderate heterogeneities in local and spatial parameters. When local and spatial dynamics exhibit high heterogeneities, however, dispersal can either stabilize or destabilize metapopulation dynamics through various mechanisms. Our findings have important theoretical and practical implications. We show that dispersal functions as a form of spatial intraspecific mutualism in metapopulation dynamics and that its effect on metapopulation stability is opposite to that of interspecific competition on local community stability. Our results also suggest that conservation corridors should be designed with appreciation of spatial heterogeneities in population dynamics in order to maximize metapopulation stability.

Introduction

One important question in spatial ecology is how spatial coupling of local populations affects the dynamics and stability of metapopulations. Natural populations undergo various sources of stochasticity and fluctuate permanently over time (Lundberg et al., 2000). A common empirical measure of stability is the temporal variability of population size (Pimm, 1984; Ives, 1995), which is closely related to the long-term persistence of the population (Inchausti & Halley, 2003). Within a metapopulation, temporal variability can be measured at both the local population and metapopulation scales. Ecological processes, particularly dispersal, may affect the variability at different scales differently (Dey & Joshi, 2006; Vogwill, Fenton & Brockhurst, 2009).

At the local scale, theoretical and empirical studies have widely documented that dispersal can decrease population variability and hence local extinction rate through spatial averaging or rescue effects (Lande, Engen & Sæther, 1998; Briggs & Hoopes, 2004; Vogwill, Fenton & Brockhurst, 2009; Abbott, 2011). However, the effect of dispersal on metapopulation stability, i.e., the stability of the whole metapopulation, has been controversial. While stabilizing local dynamics, dispersal also tends to increase the spatial synchrony between local populations (Kendall et al., 2000; Ripa, 2000; Liebhold, Koenig & Bjørnstad, 2004; Abbott, 2011). Such synchronizing effects have been demonstrated to impair the persistence and stability of metapopulations (Heino et al., 1997; Earn, Levin & Rohani, 2000; Gouhier, Guichard & Gonzalez, 2010; but see Blasius, Huppert & Stone, 1999). Therefore, dispersal is a double-edged sword (Hudson & Cattadori, 1999): it can either decrease metapopulation variability and extinction rate through its local stabilizing effects or increase them through its spatial synchronizing effects. Experimental studies have reported stabilizing, destabilizing, or no effects of dispersal on metapopulation variability (Dey & Joshi, 2006; Vogwill, Fenton & Brockhurst, 2009; Steiner et al., 2013). Given these controversies, it is necessary to use theoretical models to quantitatively study the two effects of dispersal together and understand the net effects of dispersal on stability at the metapopulation scale.

Recently, we have developed a general framework that links population or ecosystem stability across multiple scales (Wang & Loreau, 2014). When applied to a single-species metapopulation, this framework shows that metapopulation variability can be calculated as the product of local population variability and a metapopulation-wide measure of spatial synchrony. Consequently, the net effect of dispersal on metapopulation variability is determined by the relative strengths of its local stabilizing and spatial synchronizing effects. This framework provides a useful tool to investigate how the effects of dispersal (or any other factors) on variability scale up from population to metapopulation scales.

In this study, we use simple metapopulation models to investigate analytically how dispersal, in interaction with other factors, regulates temporal variability at different scales. In all our models, local populations have feasible and stable equilibria in the absence of environmental stochasticity; however, due to environmental stochasticity, populations fluctuate permanently around these equilibria. This assumption allows us to analyze our models with the linearization approximation (see Methods). We first study a homogeneous metapopulation in which local (i.e., local intrinsic growth rate and carrying capacity) and spatial (i.e., dispersal) parameters are all identical among patches. In this case, we derive analytic formulae that quantify the local stabilizing and spatial synchronizing effects of dispersal. Interestingly, we show that these two effects cancel each other out, such that dispersal has no net effect on metapopulation variability. We then consider more general cases with spatially heterogeneous population dynamics and/or asymmetric dispersal rates. Spatial heterogeneities in the environment (e.g., temperature, patch size, etc.) can result in variation in population dynamics among patches (Brown et al., 2004; Strevens & Bonsall, 2011; De Roissart, Wang & Bonte, in press). In addition, dispersal can exhibit directionality due to abiotic (e.g., water or wind flows; see Levine, 2003; Anderson, Hilker & Nisbet, 2012) or biotic (e.g., active dispersal; see Pulliam, 1988; Bowler & Benton, 2005) factors. It remains unexplored how spatial heterogeneities in population dynamics and asymmetric dispersal interact and affect the stability of metapopulations at multiple scales (but see Dey, Goswami & Joshi, 2014). By studying metapopulation models with heterogeneous local and spatial parameters, we identify several mechanisms by which dispersal can increase or decrease metapopulation variability. Finally, we discuss the implications of our model for landscape management.

Methods

Model

Consider a metapopulation composed of m local patches. The dynamics of local populations are governed by density-dependent growth and density-independent dispersal between patches. We use a continuous-time model as follows: (1) dNitdt=riNit⋅1−Nitki−diNit+∑j≠idjm−1⋅Njt+Nitεit

where Ni(t) represents the population size (or biomass) in patch i at time t, ri and ki represent the intrinsic growth rate and carrying capacity in patch i, respectively, and di represents the rate for each individual in patch i to immigrate into other patches. Here, we consider an implicit spatial structure so that individuals from patch i have equal probabilities to reach any other patch (di/(m − 1)). The random variables εi(t) represent environmental stochasticity in the growth rate of population i at time t. For simplicity, we assume that the environmental stochasticity is independent through time (i.e., white noise). The spatial correlation of the white-noise variables εi(t) are characterized by the covariance matrix Vε, for which we assume the following symmetric structure: Vε(i, i) = σ2 for any i and Vε(i, j) = ρσ2 for any i ≠ j (see Appendix S1 for details). Particularly, ρ represents the between-patch correlation in population environmental responses.

Previous studies have often used discrete-time models to study metapopulation dynamics in stochastic environments (e.g., Kendall et al., 2000; Ripa, 2000; Abbott, 2011). In these models, the order of spatial and local processes can alter the results quantitatively (Ripa, 2000). Our continuous-time model avoids this problem. In order to compare it with previous models, however, we also study two discrete-time models that capture essentially the same spatial and local processes, with distinct orderings of these processes (Appendix S3). The results under the discrete-time models are qualitatively similar to those of the continuous-time model (except for the effects of the intrinsic growth rate r; see Appendix S3). We thus present only the continuous-time model in the main text. Interested readers can find all the details about the discrete-time models in Appendix S3.

Solving for the covariance matrix

A common approach to study the temporal variability of stochastic dynamical systems is to linearize the system around its stable equilibrium (Ives, 1995; Lundberg et al., 2000; Ripa & Ives, 2003; Greenman & Benton, 2005). This linearization approach provides approximate analytic solutions for the stationary covariance matrix, on condition that the dynamical system has a stable equilibrium in the absence of stochasticity and undergoes moderate stochasticity. With this approach, we first derive the analytic solutions for homogeneous metapopulations in which local and spatial dynamics have identical parameters, i.e., for any i, ri = r, ki = k, di = d. In this case, the equilibrium local population size is simply N* = k. Around Ni(t) = N* and εi(t) = 0, Eq. (1) can be linearized into the following form (see Appendix S1): (2) dX⃗tdt=JX⃗t+N*ε⃗t

where: X⃗t=N1t−N*,…,Nmt−N*′ε⇀t=ε1t,…,εmt′J=−r−ddm−1⋯dm−1dm−1−r−d⋱⋮⋮⋱⋱dm−1dm−1⋯dm−1−r−d.

Note that J is the Jacobian matrix, and the equilibrium is locally stable when r > 0 (see Appendix S1). Under the assumption of white noise, we can derive from Eq. (2) the stationary covariance matrix of metapopulation dynamics (VN=CovX⃗∞, which is the solution of the following equation (Van Kampen, 1992; see Appendix S1): (3) VNJ′+JVN+N*2⋅Vε=0.

For general cases with heterogeneous local and spatial parameters, we analyze two-patch metapopulation models using similar procedures as above (see Appendix S2). First, by ignoring environmental fluctuations, we compute the equilibrium local population sizes (N1*,N2*) numerically. Our simulations suggest that there is always one globally stable equilibrium, i.e., trajectories starting from different initial conditions all converge to the same equilibrium. We then calculate the Jacobian matrix around this equilibrium: J=r11−2N1*k1−d1d2d1r21−2N2*k2−d2.

Based on this Jacobian matrix and the covariance matrix of population environmental responses, we can solve the stationary covariance matrix (see Appendix S2).

The covariance matrices are then used to calculate temporal variability at the population and metapopulation scales (see ‘Temporal variability at multiple scales’). To evaluate the accuracy of our linearized solutions, we also perform stochastic simulations based on the nonlinear Eq. (1). Simulation results show that the linearization generally provides a good approximation unless environmental stochasticity is large and exhibits a strong spatial asynchrony (see Fig. SA1). Note that in natural ecosystems, environmental asynchrony is not expected to be strong, because the environment generally exhibits positive spatial correlations (Moran, 1953; Hudson & Cattadori, 1999). Therefore, the linearization approach may provide a good approximation in natural ecosystems.

Temporal variability at multiple scales

Within a metapopulation, variability can be measured at alpha, beta, and gamma scales, which correspond to local population variability, spatial asynchrony among local populations, and metapopulation variability, respectively (Wang & Loreau, 2014). All these measures can be derived from the mean (N∗=N1*,N2*,…,Nm*) and covariance matrix (VN=CovX⃗∞) of local population sizes. We use the coefficient of variation (CV), i.e., the ratio of the standard deviation to the mean, to measure variability. For instance, the temporal CV of population i is: CVi=VNi,iNi*. Then, alpha variability is defined as the square of the weighted average of the local population CV: (4) αCV=∑iNi*∑jNj*⋅CVi2=∑iVNi,i∑jNj*2

and gamma variability (γcv) is defined as the square of the temporal CV of total metapopulation size: (5) γCV=∑i,jVNi,j∑jNj*2.

Beta variability or spatial asynchrony, is defined as the reciprocal of spatial population synchrony: β = 1/φp, where the spatial synchrony is defined as: (6) φp=∑i,jVNi,j∑iVNi,i2.

Metapopulation variability is then linked to local alpha variability and spatial (a) synchrony as follows: γcv = αcv ⋅ φp = αcv/β (Wang & Loreau, 2014). For the homogeneous case, we derive analytic formulae for this multi-scale variability, which is summarized in Table 1.

Table 1 Analytic solutions for multi-scale variability and spatial synchrony in homogeneous metapopulations.

For clarity, we denote d′ = md/(m − 1) and φe=1+m−1ρm. Note that by definition, we have β = αcv/γcv and φp = 1/β.

Variability or synchrony	Solution	Solution under d = 0	
Population variability (αcv)	αCV=r+d′⋅φe⋅σ22rr+d′	αCVd=0=σ22r	
Spatial asynchrony (β)	β1=r/φe+d′r+d′	β1d=0=1φe	
Spatial synchrony (φp)	φP=r+d′r/φe+d′	φPd=0=φe	
Metapopulation variability (γcv)	γCV=σ22r⋅φe	γCVd=0=σ22r⋅φe	

Dispersal-induced stability and synchrony

As demonstrated by previous studies, dispersal can simultaneously provide stabilizing and synchronizing effects on local population dynamics (Abbott, 2011). The dispersal-induced stability (Dα) can be defined as the ratio of alpha variability without dispersal to that with dispersal (Abbott, 2011): Dα=αCVd=0/αCV. Similarly, the dispersal-induced synchrony (Dφ) can be defined as the ratio of spatial synchrony with dispersal to that without dispersal: Dφ=φp/φpd=0=βd=0/β. Dα and Dφ quantify the local stabilizing and spatial synchronizing effects of dispersal, respectively. The effect of dispersal on metapopulation stability (Dγ) is determined by the relative magnitudes of these two effects, i.e., Dγ=γCVd=0/γCV=Dα/Dφ. When Dγ is larger than 1, the local stabilizing effect is larger than the spatial synchronizing effect, and thus dispersal decreases metapopulation variability. Otherwise, dispersal increases metapopulation variability.

Results

Multi-scale variability in homogeneous metapopulations

Through its stabilizing and synchronizing effects, respectively, dispersal decreases both alpha and beta variability (Fig. 1). These dispersal-induced effects are stronger under lower population growth rate, lower correlation of population environmental responses, and higher number of patches (Fig. 2). Interestingly, in homogeneous metapopulations, the dispersal-induced stability (Dα) is always identical to the dispersal-induced synchrony (Dφ); both equal the ratio of spatial population synchrony (φp) to the synchrony of environmental responses (φe=1+m−1ρm) (see Appendix S1): (7) Dα=Dφ=φpφe.

This implies that in homogeneous metapopulations, the effects of dispersal cancel out at the metapopulation level and thus dispersal has no net effects on gamma variability (Dγ = 1; see Table 1 and Fig. 1). These results, however, are based on linear approximations, which are appropriate when the environment fluctuates moderately. In a strongly fluctuating and asynchronous environment, simulations show that dispersal can provide weak stabilizing effects on gamma variability (Fig. SA1; see also Loreau, Mouquet & Gonzalez, 2003).

Figure 1 Multi-scale variability in homogeneous metapopulations.

Effects of the correlation in environmental responses (ρ), number of patches (m), and dispersal rate (d) on multi-scale variability in homogeneous metapopulations. Parameters: r = 0.5, σ2 = 0.05, and m = 10 for (A–C) and ρ = 0 for (D–F).

Figure 2 Dispersal-induced stability (Dα) or synchrony (Dφ) in homogeneous metapopulations.

Note that Dα = Dφ. Parameters for the bold line: m = 10, ρ = 0, r = 0.5, and σ2 = 0.05. Lines with marks have same parameters except: m = 5 (triangle), ρ = 0.2 (square), r = 1 (circle).

The correlation of population environmental responses (ρ) and the number of patches (m) affect the multi-scale variability mainly through their effects on the spatial synchrony of population environmental responses (φe; see Table 1). As ρ increases and/or m decreases (such that φe increases), alpha and gamma variability both increase, and the beta variability decreases (Fig. 1). Besides, as the intrinsic growth rate (r) increases, the temporal variability at alpha and gamma scales all decrease (Fig. 3). An increasing r also weakens the spatial synchronizing effects of dispersal and environmental correlation and thereby increases spatial asynchrony (Fig. 3B). Note that dispersal is required for spatial parameters (ρ and m) to affect local alpha variability and for the local parameter (r) to affect spatial asynchrony. When there is no dispersal (d = 0), alpha variability is independent of ρ and m, and spatial asynchrony is independent of r (Table 1).

Figure 3 Effects of the intrinsic population growth rate (r) on multi-scale variability in homogeneous metapopulations.

Black and red lines show results under d = 0 and 0.2, respectively. Solid and dashed lines show results under ρ = 0 and 0.2, respectively. Other parameters: m = 10, σ2 = 0.05.

Effects of spatial heterogeneities on metapopulation variability

In two-patch metapopulations, when keeping dispersal rates symmetric between the two patches, spatial heterogeneities in local parameters (r and k) generally increase gamma variability (Fig. 4). However, when the larger population (larger k) has faster local dynamics (larger r), such heterogeneity can contribute to reducing gamma variability if environmental responses are highly synchronous (Fig. 4F). When local populations have heterogeneous dynamics, increased (symmetric) dispersal rate tends to decrease gamma variability (Figs. 4A–4D). However, when local populations differ in carrying capacity (k), dispersal can be destabilizing, particularly when environmental responses are highly synchronous (Figs. 4E and 4F). Finally, note that in the cases with low or moderate heterogeneities in local parameters, symmetric dispersal has very limited effects on metapopulation variability, just as it does in homogeneous metapopulations.

Figure 4 Effect of spatial heterogeneities in local dynamical parameters and of (symmetric) dispersal rate on gamma variability in two-patch metapopulations.

(A–C): gamma variability when environmental responses are perfectly asynchronous (φe = 0); (D–F) gamma variability when environmental responses are perfectly synchronous (φe = 1). The two patches differ in their intrinsic population growth rate (r) and/or carrying capacity (k), where a larger s indicates a higher heterogeneity. Other parameters: σ2 = 0.05. See Figs. SA2 and SA3 for the patterns of variability at other scales.

When keeping local dynamical parameters (r and k) homogeneous, asymmetry in dispersal rates generally increases gamma variability (Figs. 5A and 5D). Under the extreme case where one population does not disperse, the other population will have decreased population size and increased variability as its dispersal rate increases (see Appendix S2). Consequently, the metapopulation is dominated by one population (the non-dispersing one) and thereby exhibits larger variability (Figs. 5A and 5D). However, spatial heterogeneities in local population dynamics can alter this prediction qualitatively. For instance, if the non-dispersing population has faster local dynamics (larger r), its dominance may contribute to reducing gamma variability, especially when environmental responses are highly synchronous (Fig. 5E). Moreover, when local dynamics are highly heterogeneous, the metapopulation is most stable when dispersal rates exhibit moderate asymmetries. More specifically, gamma variability is lowest when the faster population has a moderately higher dispersal rate (Figs. 5B and 5E), or when the larger population has a moderately higher dispersal rate in asynchronous environments (Fig. 5C) or has a moderately lower dispersal rate in synchronous environments (Fig. 5F).

Figure 5 Effects of asymmetric dispersal on gamma variability in two-patch metapopulations (with homogeneous/heterogeneous local dynamics).

(A–C): gamma variability when environmental responses are perfectly asynchronous (φe = 0); (D–F) gamma variability when environmental responses are perfectly synchronous (φe = 1). Symmetry in dispersal rates occurs along the 1:1 diagonal; asymmetry increases as one moves away from this diagonal. Note that along the 1:1 diagonal, gamma variability have similar patterns as those in respective panels in Fig. 4. Other parameters: σ2 = 0.05. See Figs. SA4 and SA5 for the patterns of variability at other scales.

Discussion

We have used dynamical models to study the role of dispersal, in interaction with other spatial and local factors, in regulating the stability of metapopulations at multiple scales. Both the local stabilizing and spatial synchronizing effects of dispersal have been documented in previous studies (reviewed in Abbott, 2011), and are again demonstrated by our models. One remarkable finding is that in homogeneous metapopulations, the local stabilizing effect of dispersal is always identical to its spatial synchronizing effect; consequently, dispersal has no net effect on the variability of the whole metapopulation (Fig. 1). This result is robust to moderate heterogeneities in local and spatial parameters (Figs. 4 and 5), and is consistent with findings from experiments with the same settings (i.e., experimental metapopulations with stable and homogeneous local populations; see Vogwill, Fenton & Brockhurst, 2009). In deterministic metapopulation models, previous studies have shown that random dispersal does not alter stability properties of the linearized system when local population dynamics are homogeneous (reviewed in Briggs & Hoopes, 2004). Here we have further shown that dispersal does not affect the temporal stability of homogeneous metapopulations in a fluctuating environment.

In heterogeneous metapopulations, spatial heterogeneities in local dynamical parameters or dispersal rates generally increase metapopulation variability. However, when local dynamics are heterogeneous, dispersal can provide stabilizing effects on metapopulation variability in several ways. First, by linking populations with fast and slow dynamics, dispersal can decrease gamma variability by either stabilizing both populations (Fig. SA2) or providing stronger stabilizing effects on the slower population and weaker destabilizing effects on the faster populations (Fig. SA3; see also Briggs & Hoopes, 2004; Ruokolainen et al., 2011). In particular, a moderately higher dispersal rate of the faster population can produce lowest gamma variability (Fig. SA5; see also Dey, Goswami & Joshi, 2014). Second, a much higher dispersal rate of the slower population can leave the metapopulation dominated by the faster population, which decreases gamma variability in highly synchronous environments (Fig. 5E). Third, in highly synchronous environments, while symmetric dispersal rate between small and large patches can be destabilizing (Fig. 4), a relatively higher dispersal rate of the smaller population can result in a zero net spatial flow of individuals, which decreases alpha and gamma variability (Fig. SA5). Finally, in highly asynchronous environments, dispersal can also provide stabilizing effects by reducing spatial synchrony. Specifically, a higher dispersal rate of the larger population can increase spatial evenness, which decreases spatial synchrony (Wang & Loreau, 2014) and thereby reduce gamma variability (Fig. SA4). Similarly, when asymmetries in dispersal rates operate in opposite ways—for instance the faster population has a much higher dispersal rate or a moderately lower rate, or the larger population has a higher (smaller) dispersal rate in (a)synchrounous environments—dispersal can increase the variability of the whole metapopulation.

It is interesting to compare the role of dispersal in metapopulation stability with that of interspecific competition in community stability. In metapopulations, populations interact through dispersal in physical space; in competitive communities, populations interact through interspecific competition in an abstract niche space. Remarkably, the effects of dispersal in our models are just opposite to those of competition in community stability. As shown in previous studies, competition can increase species variability but simultaneously decrease species synchrony; in symmetric communities with identical species parameters, these two effects cancel each other out and consequently competition has no effect on community stability (Hughes & Roughgarden, 1998; Ives, Gross & Klug, 1999; Loreau & De Mazancourt, 2008; Loreau & De Mazancourt, 2013). In asymmetric communities, an increasing asymmetry in competitive abilities generally increases community variability (Hughes & Roughgarden, 1998; Loreau & De Mazancourt, 2013); however, a moderately higher competitive ability of the slower species, or a much higher competitive ability of the faster species, can decrease community variability (Loreau & De Mazancourt, 2013). All these effects demonstrate the opposite roles played by interspecific competition in community stability and dispersal in metapopulation stability. These contrasting patterns could be understood from the fact that competition reduces the population size of recipient species while dispersal increases the size of recipient populations. This is reflected in the Jacobian matrices in which competition and dispersal produce negative and positive interaction coefficients, respectively. In other words, dispersal acts as a form of spatial intraspecific mutualism in the dynamics of metapopulations.

Our results have important implications for landscape management. Corridors are commonly promoted as a conservation strategy to mitigate the effects of habitat fragmentation. Corridors have been demonstrated to promote dispersal and movement between habitat patches (Gilbert-Norton et al., 2010). But evidence is still lacking about the effects of corridors on population persistence (Haddad et al., 2011; but see Gonzalez et al., 1998), which is the ultimate reason for creating corridors. Our results suggest that corridors do not necessarily increase the stability and persistence of metapopulations (see also Earn, Levin & Rohani, 2000). In the specific case where local populations have nearly identical dynamical parameters, the synchronizing effect of dispersal cancels out their local stabilizing effect such that corridors have no net effect on the stability at metapopulation level. In reality, spatial heterogeneity can result in a stronger or weaker local stabilizing effect of dispersal compared to its synchronizing effect, and hence corridors may enhance or impair the stability of metapopulations. Our model suggests that in a heterogeneous landscape, the most efficient design of corridors is often one that generates asymmetric dispersal (Fig. 5). For instance, if local patches have different growth rates, the metapopulation is most stable when the faster-growing patch has a moderately higher dispersal rate than the slower-growing patch. Such asymmetric dispersal might be achieved by two mechanisms. First, when connected by corridors, the faster-growing population may have a higher level of dispersal activity spontaneously, e.g., by active dispersal (Pulliam, 1988). Second, it might be possible to design corridors that produce directional dispersal in some cases, for instance by taking advantage of water and/or wind flow (Säumel & Kowarik, 2010; Anderson, Hilker & Nisbet, 2012).

Concluding remarks

In this paper, we have explored how spatial processes govern the variability of metapopulations at multiple scales in a stochastic environment. We show that within a metapopulation, dispersal functions as a form of spatial intraspecific mutualism. While stabilizing local populations, dispersal has very limited stabilizing effects on metapopulations if local population dynamics are homogeneous. In highly heterogeneous metapopulations, however, dispersal can stabilize or destabilize metapopulations through various mechanisms. Therefore, corridor designs, in order to increase metapopulation stability and persistence, should be context dependent with explicit consideration of spatial heterogeneities in population dynamics.

Our results are based on a simple metapopulation model and thus its limitations should be kept in mind. First, in our models local populations always have stable equilibrium if there is no environmental stochasticity. However, if local populations undergo complex dynamics (e.g., limit cycles or chaotic attractors), dispersal can provide stabilizing effects through interacting with nonlinearity and spatial heterogeneity (Briggs & Hoopes, 2004; Abrams & Ruokolainen, 2011; Dey, Goswami & Joshi, 2014; see also Fig. SA1). Our model also ignores the effects of environmental autocorrelation (Ruokolainen et al., 2009). We have shown that our first main result, i.e., the stability of homogeneous metapopulations is not affected by dispersal, still holds for coloured noise (see Appendix S4). Still, it would be worthwhile to investigate the interactive effects of coloured noise and spatial heterogeneities on the dispersal-stability relationship in future research. Besides, our model ignores the effects of interspecific interactions, the inclusion of which may alter some of our conclusions (Koelle & Vandermeer, 2005). For instance, corridors can promote species co-occurrence and thus enhance competition or predation pressure, which thereby may impair the persistence of the focal species (Loreau, Mouquet & Gonzalez, 2003; Vogwill, Fenton & Brockhurst, 2009). Finally, our model considers space implicitly, which could be extended to a spatially explicit one to study the interactive effects between dispersal, spatial heterogeneity, and landscape configuration (Holland & Hastings, 2008). Future studies should incorporate these complexities to better understand the effects of dispersal on patchy populations in stochastic environments.

Supplemental Information

Supplemental Information 1 Supplementary appendices

Appendix 1, Continuous-time models and their analytic solutions in homogeneous metapopulations. Appendix 2, Continuous-time models with spatial heterogeneity. Appendix 3, Discrete-time models and their analytic solutions in homogeneous metapopulations. Appendix 4, Environmental stochasticity beyond white noise.

Click here for additional data file.

Figure SA1 The effect of dispersal and the variance of environmental stochasticity on the gamma variability of homogeneous metacommunities

The effect of dispersal and the variance of environmental stochasticity on the gamma variability of homogeneous metacommunities, based on stochastic simulations (A, C, E) and analytic solutions from linear approximations (B, D, F). Note that (A) and (B) are of same scale, (C) and (D) are of same scale, and (E) and (F) are of same scale. Parameters: m = 2, r = 0.5, σ2 in [0.01, 0.5], d in [0, 1], and ρ = − 0.9, 0 or 0.9.

Click here for additional data file.

Figure SA2 Effect of spatial heterogeneities in local dynamical parameters and of (symmetric) dispersal rate on the multi-scale variability

Effect of spatial heterogeneities in local dynamical parameters and of (symmetric) dispersal rate on the multi-scale variability of two-patch metapopulations when environmental responses are perfectly asynchronous (φe = 0). The two patches differ in their intrinsic population growth rate (r) and/or carrying capacity (k), where a larger s indicates a higher heterogeneity. Note that the patterns of gamma variability (γcv) have been shown in Figs. 4A–4C.

Click here for additional data file.

Figure SA3 Effect of spatial heterogeneities in local dynamical parameters and of (symmetric) dispersal rate on the multi-scale variability

Effect of spatial heterogeneities in local dynamical parameters and of (symmetric) dispersal rate on the multi-scale variability of two-patch metapopulations when environmental responses are perfectly synchronous (φe = 1). The two patches differ in their intrinsic population growth rate (r) and/or carrying capacity (k), where a larger s indicates a higher heterogeneity. Note that the patterns of gamma variability (γcv) have been shown in Figs. 4D–4F.

Click here for additional data file.

Figure SA4 Effect of symmetric dispersal on the multi-scale variability in two-patch metapopulations

Effect of symmetric dispersal on the multi-scale variability in two-patch metapopulations (with homogeneous/heterogeneous local dynamics) when environmental responses are perfectly asynchronous (φe = 0). Note that the patterns of gamma variability (γcv) have been shown in (Figs. 5A–5C).

Click here for additional data file.

Figure SA5 Effect of asymmetric dispersal on the multi-scale variability in two-patch metapopulations

Effect of asymmetric dispersal on the multi-scale variability in two-patch metapopulations (with homogeneous/heterogeneous local dynamics) when environmental responses are perfectly synchronous (φe = 1). Note that the patterns of gamma variability (γcv) have been shown in (Figs. 5D–5F).

Click here for additional data file.

We thank Claire de Mazancourt for discussions, and Dries Bonte, Elad Shtilerman, and anonymous reviewers for helpful comments on earlier versions of the manuscript.

Additional Information and Declarations

Competing Interests

Author Contributions

The authors declare there are no competing interests.

Shaopeng Wang conceived and designed the experiments, performed the experiments, wrote the paper.

Bart Haegeman performed the experiments, wrote the paper.

Michel Loreau conceived and designed the experiments, wrote the paper.

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
