# Peer review of "Dispersal and metapopulation stability"

_PeerJ, doi:10.7717/peerj.1295_

## Round 0.1 · original submission · Minor Revisions

This is a very nice paper, but there are still some issues that need to be addressed. I completely agree with reviewer 1's comments about the use of the word symmetric -- it should be replaced. I also completely agree with the comments of reviewer 2 indicating ways to increase the accessibility of the manuscript. I look forward to your revision.

·

Basic reporting

Generally written well. One issue which is confusing is the choice of the words "symmetric"/"symmetry"/"asymmetry" (for example in line 75): in this MS the term is used to describe a homogenous metapopulation (symmetric) - i.e. a metapopulation in which the parameters for all the local populations are identical or a heterogeneous metapopulation (asymmetric). Because there exist several studies (eg. Shtilerman (2015) and references therein) in which the term symmetry is used for dispersal network symmetry, meaning that a dispersal route from A to B implies the existence of a route from B to A, I suggest to replace this word with "homogenous"/"heterogeneous" or with "uniform"/"non-uniform".
other very minor issues are:
1) line 72 - "interactions" -> "interaction"
2) line 75 - "equilibriums" -> "equilibria"
3) line 233 appendix 3 - "... of the both side ..." -> "... of both sides of ..."



Shtilerman, E., & Stone, L. (2015). The effects of connectivity on metapopulation persistence: network symmetry and degree correlations. Proceedings of the Royal Society of London B: Biological Sciences, 282(1806), 20150203.

Experimental design

It seems to me that the MS can benefit from a generalization of the asymmetric case, specifically it would be interesting if a system of more than two patches was studied. Perhaps this can be accomplished through a numerical procedure instead of an analytic approach. If this isn't possible perhaps this choice should be better justified.

Validity of the findings

See about the experimental design - The validity of the section about asymmetric dispersal will be boosted by examining a model with a number of patches greater than two.

Reviewer 2 ·

Basic reporting

I thoroughly enjoyed reading this article. The structure of the text followed a clear path and the methods, results and figures were appropriate for the content of the piece. I do however, have a few suggestions that the authors may incorporate.

The introduction feels a bit to short for my liking - and I believe a bit more context and background would really help to motivate this work. First, there is a fairly broad literature about the importance of dispersal on other aspects of metapopulation stability (e.g. extinction risk), both theoretical and empirical, which could be added to the text to round it out a bit more completely. Second, the role of network topology has been fairly well discussed in this literature, but is really lacking here. I think the authors should introduce the point about topology being important, and re-raise the issue in the discussion where their findings might be extended to topologies where dispersal isn't global across the entire metapopulation.

Additionally, I was a bit uneasy with the generality with which the authors applied their findings, until I read Appendix 1. And I suspect that other readers may have a similar reaction to mine. My uneasiness was based around the idea of using a linearization about the equilibrium to solve the variance-covariance structure of the metaopopulation - where the variation is driven by a stochastic process in time and space. My first reaction to this was that the results would be entirely dependent on the time scale at which the "switches' in the environmental states were incorporated so I was a bit suspicious that the analytical results would really be relevant in this case. Appendix 1 does a very nice job of selling this aspect of the method and I'd argue that a paragraph or two in the main text, about the validity of derivation of equation 3 would be quite valuable.

Often the word "equilibriums" is used, but more oft used plural is "equilibria ".

I found the wording referring to "local" and "spatial" parameters confusing. It is unclear in this paper what parameters constitute the local and spatial parts of the model. Most metapopulation studies which use different parameters to govern density dependence in different localities typically refer to it as "demographic heterogeneity" and this wording would likely help to clarify the text on page 4 here as well.

I'd recommending numbering equations for alpha, beta and gamma variability on page 8 so that they can be more easily referenced by the reader - perhaps as a single group if space is an issue.

Experimental design

The design is very rigorous and uses a model structure that is general and commonly used in this area of work. I have one concern about the design, as it relates to the generality of the findings, which is described below.

Validity of the findings

I believe there is still one issue that the author should discuss more fully about their design as it relates to the generality of the findings. The linearization about the equilibrium used to determine the Jacobian is asymptotically biased so that the larger the variance of e(t) becomes, the less useful the analytical results, because the nonlinearity of the model will begin to impact the covariances of population dynamics in manners that are not caught by the present analysis. I think it would be a useful addition to discuss how analytical and stochastic solutions diverge as the variance of e(t) increases - perhaps an additional figure.

Additional comments

As I stated above, I really enjoyed reading this article. I'm not sure that most readers really appreciate the puzzle that spatial synchronization poses, given the common conception and the amount of theoretical support for the idea that synchronization is a 'bad' thing in metapopulations. This point could be strengthened in the intro and discussion to really enhance the impact of this work.

---

## Round 0.2 · accepted · Accept

Thanks for the careful revision.